# Tongue Strength of Older Adults Requiring Long-Term Care Varies throughout the Day

**DOI:** 10.3390/geriatrics8060107

**Published:** 2023-10-26

**Authors:** Shin Yoshinaka, Kohei Yamaguchi, Ryosuke Yanagida, Miki Ishii, Kazuharu Nakagawa, Kanako Yoshimi, Ayako Nakane, Yusuke Matsuyama, Jun Aida, Haruka Tohara

**Affiliations:** 1Department of Dysphagia Rehabilitation, Graduate School of Medical and Dental Sciences, Tokyo Medical and Dental University (TMDU), 1-5-45 Yushima, Bunkyo-ku, Tokyo 113-8510, Japan; slotdds4216@gmail.com (S.Y.); ry.yanagida@gmail.com (R.Y.); mickey.cookie05015@gmail.com (M.I.); nakagerd@tmd.ac.jp (K.N.); k.yoshimi.gerd@tmd.ac.jp (K.Y.); a.nakane.swal@tmd.ac.jp (A.N.); harukatohara@hotmail.com (H.T.); 2Department of Oral Health Promotion, Graduate School of Medical and Dental Sciences, Tokyo Medical and Dental University (TMDU), 1-5-45 Yushima, Bunkyo-ku, Tokyo 113-8510, Japan; matsuyama.ohp@tmd.ac.jp (Y.M.); aida.ohp@tmd.ac.jp (J.A.)

**Keywords:** tongue strength, dysphagia, older adults, oral mucosal moisture, hand grip strength

## Abstract

Physical performance shows approximately 30% diurnal variation; however, diurnal variation in oral function remains unclear. This study aimed to determine the diurnal variation in oral and swallowing function in older adults requiring long-term care. The participants included 13 adults aged >60 years (3 men and 10 women, mean age: 77.2 ± 6.3 years, age range: 62–90 years) requiring long-term care. Tongue strength (TS) and oral mucosal moisture were measured as indices of oral and swallowing function, while hand grip strength was measured as an index of general muscle strength. The patients were asked to participate in a “test” after breakfast, lunch, and dinner on the same day. Multilevel linear regression analysis was used to examine diurnal differences in each item. Multilevel linear regression analysis with adjustment for age and sex revealed that TS was significantly higher at noon (*p* = 0.001) than in the morning. Therefore, caregivers who provide support during meals to older people requiring long-term care should consider the possibility of swallowing function differing according to the time of the day. In conclusion, it may be beneficial to establish a nutritional therapy that accounts for the diurnal variation in TS.

## 1. Introduction

It remains unclear whether the circadian rhythm applies to oral and swallowing functions. Mammals have a circadian rhythm of ≈24 h per day. The circadian rhythm is controlled by clock genes such as *Clock* and *Bmal1* [1]. Older adults tend to go to bed and wake up early based on the expression of clock genes [2]. Moreover, clock genes affect body temperature, blood pressure, and hormone secretion, and are associated with various diseases [3,4].

Approximately half of older adults requiring long-term care present with dysphagia [5], which can lead to aspiration pneumonia and malnutrition [6]. The tongue is an important organ responsible for mastication, swallowing, and articulation. Tongue strength (TS) is an easily measurable factor [7] that is associated with dysphagia and undernutrition [8]. Systematic reviews have reported that low TS affects clinical outcomes, including poor swallowing recovery in hospitalized patients, increased incidence of pneumonia, and poor life expectancy [9]. Moreover, TS is associated with whole-body muscle strength [10]. Previous studies have reported a relationship between TS and physical performance, and treadmill exercises have been shown to improve tongue endurance and tongue muscle properties in rats [11].

Additionally, physical rehabilitation and nutritional management have been shown to improve TS in patients with sarcopenia in convalescent hospitals [12]. Furthermore, improving physical function has been shown to improve dysphagia through increased TS in patients with acute heart failure [13]. Physical performance shows diurnal variations of up to 26% [14]. One of the factors contributing to the circadian variation in physical performance is the increase in body temperature during the day, which in turn improves blood flow and increases nerve transmission speed [2]. 

Chewing and swallowing are closely related [15]; additionally, the tongue and saliva are crucially involved in food mass formation while chewing. There are diurnal variations in the amount of saliva secretion [16]; however, it remains unclear whether the same tendency exists in older adults who require nursing care. Older adults requiring long-term care may show diurnal variations in TS, oral lubrication, and physical function. Nutritional therapy may be tailored to account for diurnal variations in swallowing function, such as increasing nutritional intake during times of high swallowing function.

However, to the best of our knowledge, no study has evaluated the diurnal variation in oral and swallowing functions among older adults requiring long-term care. Elucidating high-risk periods for dysphagia may inform the schedules for caregivers. Therefore, this study aimed to clarify the diurnal variation in TS and oral mucosal moisture as indexes for oral and swallowing functions as well as to clarify the diurnal variation in hand grip strength (HGS) as an index for assessing systemic muscle strength in the morning, noon, and evening. Based on a previous study [14], we hypothesized that TS, oral mucosal moisture, and HGS in older adults requiring long-term care would be higher at noon and in the evening.

## 2. Materials and Methods

### 2.1. Participants

We included individuals aged >60 years living at either of two long-term healthcare facilities. The inclusion criteria were as follows: (i) ability to follow instructions, (ii) ability to take meals orally, and (iii) having any systemic diseases in the chronic phase. The exclusion criteria were as follows: (i) severe dementia or extreme difficulty following instructions, (ii) obvious paralysis of the tongue or upper extremities, (iii) reliance on tubal feeding, and (iv) reliance on medical care due to acute diseases or extreme malnutrition. Accordingly, we included 13 participants (3 men and 10 women; mean age ± standard deviation [SD], 77.2 ± 6.3 years; age range, 62–90 years). Among them, one participant became agitated during oral mucosal moisture and TS evaluation at noon, which impeded the assessment.

This study was conducted according to the Declaration of Helsinki of 1964, revised in 2013, and was approved by the Ethics Committee of the Faculty of Dentistry, Tokyo Medical and Dental University (No. D2021-075). Participants or their legal representatives were provided with sufficient verbal and written explanations regarding this study. Subsequently, written informed consent was obtained from all participants or their legal representatives.

#### Collection of Participant Data

Data regarding age, sex, body mass index (BMI), frailty level, and cognitive function were collected from the medical records of the long-term healthcare facilities. The clinical frailty scale was used as an index of frailty [17]. Cognitive function was assessed using the Mini Mental State Examination (MMSE) and Revised Hasegawa Dementia Scale (HDS-R).

### 2.2. Procedure

The measurements were performed by a dentist and speech therapist with >20 years of experience in evaluating dysphagia. Within 1 day, the dentist visited one long-term healthcare facility thrice and measured TS, HGS, and oral mucosal moisture; accordingly, measurements in both facilities were completed in 2 days. Regarding mealtimes at the facilities, breakfast, lunch, and dinner were served at 7:30/8:00 a.m., 12:00 p.m., and 6:00 p.m., respectively. The measurements were performed after each meal, i.e., 7:30 a.m.–9:00 a.m., 12:00 p.m.–1:00 p.m., and 6:00 p.m.–7:00 p.m.

### 2.3. TS, HGS, and Oral Mucosal Moisture

TS was measured using a JMS TS device (JMS Co., Ltd., Hiroshima, Japan). In a sitting position, the participants were asked to place the balloon in their mouths and hold the plastic pipe with their upper and lower central incisors with their lips closed (Figure 1). 

The dentist held the probe against the middle of the tongue while recording the measurement. Subsequently, the participants were asked to push the balloon with the middle of their tongue against the hard palate for ≈7 s with maximum strength. Based on two previous studies [7,18], the TS measurement was performed three times, and the average value was recorded. We have previously reported high inter-rater reliability for TS measurements [19].

HGS was measured using a digital grip dynamometer (Takei Scientific Instruments Co., Ltd., Niigata, Japan) with the participants in a sitting position. The participants were asked to squeeze the handle using their maximum strength. The HGS measurement was continued until the numerical value on the measuring instrument stabilized. Further, HGS was measured twice with the dominant hand, and the higher score was recorded.

Oral mucosal moisture was measured using an oral moisture-checking device, “Mucus” (Life Co., Ltd., Saitama, Japan) (Figure 1). Based on a previous study [20], the device was covered with a disposable polyethylene sensor cover. Mucosal moisture was measured in the central region of the tongue mucosa at 10 mm from the tongue tip. The device was pressed to the tongue mucosa with a pressure of 200 gf. The measurement was performed three times at 15 s intervals with the participants’ mouths closed. The mean value of the three measurements was recorded.

### 2.4. Frailty

Frailty was evaluated using the clinical frailty scale [17], which is scored from 1 (very fit) to 9 (terminally ill) based on the severity of frailty. Scores of 6 and 7 indicate moderate frailty requiring nursing care and complete nursing care, respectively; however, the 6-month mortality risk remains low.

### 2.5. Cognitive Function

#### 2.5.1. MMSE

The MMSE is a screening test for assessing cognitive function. It comprises 11 items including time orientation, place orientation, 3-word immediate and delayed play back, calculations, item designations, sentence repetition, 3-level verbal commands, writing instructions, sentence writing, and figure copying. An MMSE score ≤ 23 is suggestive of dementia [21], while a score ≤ 27 suggests mild cognitive impairment [22].

#### 2.5.2. HDS-R

The HDS-R is another screening test for assessing cognitive function. It consists of nine items: age, orientation, immediate three-word memorization and delayed recall, arithmetic, number reversal, object memorization, and verbal fluency, with a maximum score of 30. The cutoff HDS-R score for suspicion of dementia is 20 [23].

### 2.6. Statistics

Age, BMI, TS, oral mucosal moisture, HGS, and cognitive function are presented as the mean and SD, and the clinical frailty scale shows the distribution by scores. TS, oral mucosal moisture, and HGS are presented as median (interquartile range) values. We evaluated the differences in TS, oral mucosal moisture, and HGS according to the time of day (i.e., morning, noon, and evening) with adjustment for age and sex through a multilevel linear regression analysis. A multilevel model was employed to account for the hierarchical data structure, wherein each observation—at morning, noon, and evening—is nested within individuals.

## 3. Results

The mean BMI of the 13 study participants was 21.5 ± 2.5 kg/m^2^. All participants had a clinical frailty scale score ≥ 6, with 69.2% (9/13), 23.1% (3/13), and 7.7% (1/13) of the participants having scores of 6, 7, and 8, respectively. Cognitive function was evaluated using the MMSE (n = 3, 17.33 ± 0.94) and HDS-R (n = 4, 9.75 ± 4.76), with six participants not being evaluated. The scores for the seven participants who underwent assessments were below the respective cutoff values.

The mean values ± SD of TS were 19.3 ± 7.7, 22.6 ± 9.1, and 22.4 ± 8.3 kPa in the morning, noon, and evening, respectively. Additionally, the mean values ± SD of HGS were 13.8 ± 7.6, 14.2 ± 7.6, and 13.4 ± 7.5 Kg in the morning, noon, and evening, respectively. The values for oral mucosal moisture were 27.4 ± 3.9, 28.4 ± 2.1, and 27.6 ± 2.2 in the morning, noon, and evening, respectively. 

The median (interquartile range (IQR)) values of TS were 19.6 (6.1), 20.7 (3.2), and 21.9 (3.6) kPa in the morning, noon, and evening, respectively. Further, the median (IQR) values of HGS were 11.0 (0.5), 12.1 (3.0), and 11.3 (1.5) Kg in the morning, noon, and evening, respectively. The values for oral mucosal moisture were 28.4 (1.1), 29.0 (0.7), and 28.5 (3.0) in the morning, noon, and evening, respectively (Figure 2). 

Table 1 shows the results of the multilevel linear regression analysis. TS at noon was significantly higher than in the morning (*p* < 0.001); however, there was no significant difference between the values obtained in the morning and evening (*p* = 0.059). Additionally, there was no significant difference in HGS or oral mucosal moisture at any time.

## 4. Discussion

This study investigated the diurnal variation in TS, oral mucosal moisture, and HGS in older adults requiring long-term care in nursing facilities. We found that TS was significantly higher at noon than in the morning; however, there were no significant differences in oral mucosal moisture and HGS throughout the day.

As we age, the biological clock changes by shortening and advancing its cycle. The rhythms of clock gene expression also decrease, and the related physiological functions decrease in accordance with the amplitude of circadian rhythms. Peak performance times for college students vary by chronotype, with morning students having a peak performance at 12:12 a.m. Additionally, sports performance shows diurnal variation of up to 26% [14]. Given the age-related changes in biological clocks, the peak performance times for older adults are expected to be advanced and have less diurnal variation. In the present study, TS was 17.1% higher after lunch (11:00 am–1:00 pm) than after breakfast (7:00 am–9:00 am). Further, there was a relatively small difference in the TS values between morning and noon (3.3 kPa). Notably, the mean TS values in the morning and at noon were 19.3 and 22.6 kPa, respectively, which may influence the diagnosis of sarcopenic dysphagia since the cutoff TS value for sarcopenic dysphagia is 20.0 kPa [24]. Accordingly, it may be important to consider diurnal variations in TS in clinical settings.

Diurnal changes in body temperature are among the observed circadian variations of bodily functions [25]. In humans, the body temperature changes by about 1 degree during the day; further, older people show an earlier peak time of body temperature (≈21:00) than adults [2]. The nerve conduction velocity slows down when the body temperature drops. A previous study on 25 swimmers measured circadian rhythms in body temperature and swimming performance eight times at 3-h intervals within a day and observed a significant association between them [26]. However, the diurnal variation in body temperature cannot sufficiently explain diurnal variation in TS by itself. The tongue is involved in various functions, such as mastication and swallowing. Compared with the skeletal muscles of the extremities, the tongue has an extremely large range of motion; moreover, it is anatomically complex and unique. It is important to investigate the circadian change in TS with respect to the effects unique to the tongue, with respiration being among the important factors. Respiration activates the tongue [27], and TS and tongue endurance in older adults with higher physical activity have been reported to be significantly higher than those in young adults with lower physical activity [28]. Further, TS has been associated with engagement in leisure activities [29]. The activity level of older people has been reported to peak at 11:00 am [30]. Taken together, TS may peak between 11:00 am and 1:00 pm since respiration increases with physical activity until around noon. 

Diurnal variation in TS and HGS could differ. In this study, TS tended to differ across the day, with significant differences between the values obtained in the morning and at noon. Contrastingly, HGS showed no significant differences across the day. Given the anatomical differences between the tongue and limb muscles, the effects of the biological clock on respective muscles may differ. Approximately 60% of the tongue and upper comprise type 1 [31] and type 2 fibers [32], respectively. Type 1 muscle fibers undergo atrophy by disuse [33], while type 2 fibers undergo atrophy with aging [34]. *Atrogin1* is one of the genes involved in muscle hypertrophy; however, its expression shows diurnal variation and is associated with skeletal muscle atrophy [35]. Accordingly, age-related changes in the biological clock may have a greater effect on limb muscles than on the tongue. Although some cross-sectional studies have reported a relationship between TS and HGS [10], the differences in diurnal variation must be evaluated separately for TS and HGS. Owing to the small sample size of this study, the possibility of beta error cannot be ruled out, as well as the fact that no significant differences were found in HGS. Future studies including a larger number of participants will be necessary.

In a previous study of 28 people in their 20s to 40s and 60s to 80s, no age-related changes were observed in salivary flow either at rest or during stimulation, and the saliva volume remained stable after 1 and 2 h [36]. In our study targeting older adults requiring long-term care, the degree of oral mucosal moisture was measured after meals, and the results were consistent with those of the previous study [36].

This study has several limitations. First, we included a small sample size. Accordingly, a large-scale study accounting for various confounding factors is warranted. Second, we did not investigate several factors that are related to TS, including the amount of activity during the day, consumption of a texture-modified diet, and frequency of verbal communication warranted to elucidate the diurnal differences in swallowing function and systemic muscle strength among older adults. Third, measurements were taken in the morning, noon, and evening on the same day; therefore, we could not exclude the influence of learning effects, especially on TS measurement [37]. However, the TS value was not the highest in the evening (the timing when the maximum proficiency of the measurement method was expected), which indicated an insignificant influence of the learning effect on the findings. In this study, the measurement of TS took approximately 7 s, consistent with that in previous studies [7,18]. We used an average of three measurements. However, there is potential for tongue fatigue to set in, which may affect the results. Additionally, a discrepancy was noted between the maximum value of TS and our measurements.

Despite its limitations, this study has important clinical implications. TS varies throughout the day; it is lower in the morning and higher at noon. There is a great deal of knowledge about “what” and “how” to eat, such as a texture-modified diet [38] and resistance training for swallowing-related muscles [39]. However, the concept of time, i.e., when to eat what, has not been elaborately incorporated in current dysphagia rehabilitation. This study provides important basic findings for establishing updated dysphagia rehabilitation that incorporates the concept of time.

## 5. Conclusions

In conclusion, TS around noon is significantly higher than during the morning. It is necessary for caregivers who provide support during meals to older people requiring nursing care to consider the diurnal variations in swallowing function.

Further large-scale studies are warranted to investigate the diurnal differences in oral, swallowing, and physical functions, which could inform novel methods for nutritional intervention and dysphagia rehabilitation guided by the concept of time.

## Figures and Tables

**Figure 1 geriatrics-08-00107-f001:**
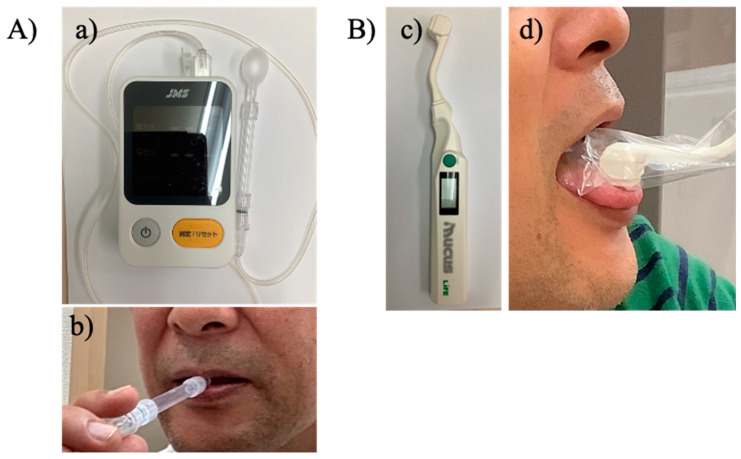
Measurement of tongue strength and oral mucosal moisture. (**A**) Tongue strength. (**a**) Tongue strength measurement device (**b**) Tongue strength measurement. (**B**) Oral mucosal moisture. (**c**) Oral moisture-checking device “Mucus” (**d**) Oral mucosal moisture measurement.

**Figure 2 geriatrics-08-00107-f002:**
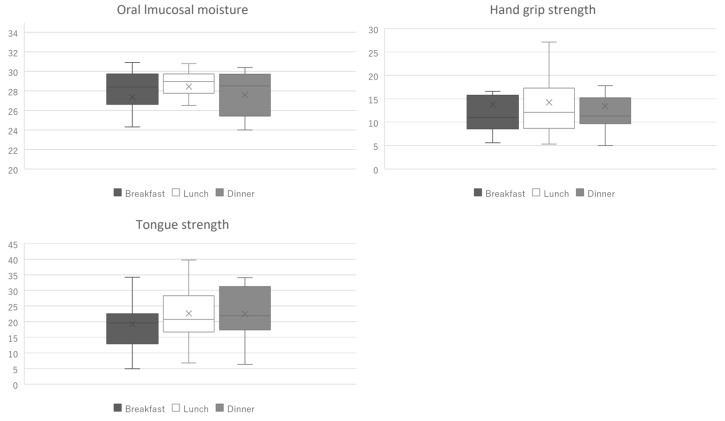
The box plot of oral mucosal moisture, hand grip strength, and tongue strength.

**Table 1 geriatrics-08-00107-t001:** Association of the inspection time with tongue strength, oral mucosal moisture, and hand grip strength adjusted for age and sex by multilevel linear regression.

Factor	Category	Oral Mucosal Moisture (n = 12 Individuals)	Hand Grip Strength (Kg) (n = 13 Individuals)	Tongue Strength (kPa) (n = 12 Individuals)
		Coefficient (95% CI)	Coefficient (95% CI)	Coefficient (95% CI)
Time	morning	Reference	Reference	Reference
	noon	1.083 (−1.764–3.930)	0.438 (−1.122–1.999)	3.065 (1.257–4.873) *
	evening	0.225 (−1.934–2.384)	−0.362 (−2.122–1.399)	2.600 (−0.095–5.295)
Sex	women	Reference	Reference	Reference
	men	−1.896 (−3.399–−0.392) *	11.284 (1.843–20.725) *	4.017 (−7.145–15.178)
Age		−0.135 (−0.268–0.001) *	−0.241 (−0.491–0.008)	−0.349 (−0.846–0.148)

* *p*-value < 0.05.

## Data Availability

The data that support the findings of this study are available from the corresponding author, K.Y., upon reasonable request.

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
