# Peer review of "Tongue Strength of Older Adults Requiring Long-Term Care Varies throughout the Day"

_geriatrics, 2023, doi:10.3390/geriatrics8060107_

Round 1

Reviewer 1 Report

Comments and Suggestions for Authors

This is a well-written paper with an interesting premise. Unfortunately, the approach for determining maximum performance for tongue strength appears to be invalid and the analyses provide contradictory and uninterpretable results. The implications and conclusions are overreaching based on the methods and low-power of the data.

Abstract.

The statement in the abstract that TS was significantly higher at noon is immediately followed by the statement that no significant differences were found.  Please correct this inconsistency. 

TS is not a clear indicator of the need or level of modified diets.  This is an inappropriate conclusion in the abstract and also at the end of the discussion.

Background

The idea of circadian rhythms in human performance is interesting, but it seems much more important to focus on variability for these measures in general.  There are studies that have examined day-to-day variability in tongue strength measures. For example, O’Day et al. (https://doi.org/https://doi.org/10.52010/ijom.2005.31.1.2) studied tongue strength and hand-grip strength in older adults across 4 days.  Their results show a learning effect for the tongue measures that may also play a role in the present study.

2.1. Inclusion criteria: iv. “obvious paralysis” – would this include obvious paresis?

2.2. Did the speech therapist and dentist each have > 20 years of experience or is that number collective?

Why was the TS maneuver sustained for 7 seconds? Is the response time of the JMS TS device that slow?  Typically, TS is measured in 1-2 seconds.  This allows for multiple measures without inducing fatigue.

If the goal is to determine maximum TS, then why isn’t the maximum value used? Averaging maximum voluntary contractions (MVCs) over 7 seconds, and then averaging those results over 3 trials is sure to provide depressed values. Although this method was used in two previous studies, I fail to understand how this could be a valid procedure for determining maximal performance. (Disclaimer: I did not read the previous studies to check for proof of validation of the procedures.)

The authors followed accepted procedures for determining strength for HGS (one MVC squeeze, best of 2 trials – although best of 3 trials is usually considered best practice).  Why wasn’t this method used for TS?

2.3. Using the average of three measurements is appropriate for this measure.

2.4. I question the use of the Kruskal-Wallis test for the comparisons. I assume that the data did not meet the necessary assumptions for a repeated-measures multivariate analysis of variance, but this nonparametric test does not seem adequate for this purpose.

3. Results. Table 1 is mentioned on line 135, but this does not match the Table 1 provided in the document. According to the text, this table should list demographic information about the participants.

Age: If all participants were >70 years of age, then why state they were > 65 years in line 130.  Since this is the results section, you can simply state their actual ages, not what was planned. What was the age range of participants?

If one person couldn’t produce TS data at noon, then the main result from this paper is due to only 12 participants. I recommend removing the 13th participant from all analyses, not just the two data points that are missing and stating clearly in the abstract and results that 12 participants completed all tasks for this study and contributed to data analysis.

Table 1. I don’t understand this table. The first two columns need headings (Factor? Time of Day?)

“The median (IQRs) for TS were 20.8 (8.6), 20.7 (7.8), and 20.0 (10.8 ) in the morning, noon, and evening, respectively.” (Lines 140-141) [Note: please add units to your numbers.] So, how is it possible that TS at noon is significantly different according to the regression analysis?  There is clearly something wrong with the analysis.  Please show a graph of the data (perhaps a box-and-whisker plot) or a table of descriptive statistics so that readers can determine for themselves how confident they can be with the results.

Discussion.  The entire discussion is focused on a significant increase in TS at noon, whereas that does not seem to the be case.  The results are grossly overinterpreted and the discussion is misleading.

The information in the discussion is scholarly, interesting, and well written.  But it is far removed from the data. The discussion (lines 201-217) about dysphagia is irrelevant and unnecessary. There are many issues involved in dysphagia, and tongue strength is not the most direct measurement for predicting dysphagia. The idea that tongue weakness predicts dysphagia is grossly oversimplified and can mislead readers. For example, on lines 225-228, the authors suggest that reduced TS can increase aspiration risk without dealing with the root causes of aspiration risk (e.g., reduced sensation, reduced hyolaryngeal elevation).  Does sensory function vary throughout the day?  Could results be related to somnolence, attention, posture, or something else? In fact, as far as I can tell, there is no time-of-day difference in the data, so none of these issues are relevant except to inform clinicians that they can measure TS accurately any time of day.

Reviewer 2 Report

Comments and Suggestions for Authors

Thank you for sharing this manuscript with me. It is great to see efforts taken to clarify the diurnal variation in TS and oral mucosal moisture as indexes for oral and swallowing functions and to clarify the diurnal variation in hand grip strength (HGS) as an index for assessing systemic muscle strength in the morning, noon, and evening. However, I would like clarification on some points:

Introduction

Ÿ   Line 21: TS was significantly higher at noon (p = 0.001) than in the morning, and no significant differences were found. TS is significantly higher than in the morning and the P value showed a significant difference. Why do authors claim that no significant differences were found?

Ÿ   Line 48-50: Regarding “In the older adults requiring long-term care, there appears to be a distinct diurnal variation in tongue pressure and oral wetting level, as well as physical function.” the author mentioned. Where does this argument come from? Suggest adding reference citations.

Ÿ   Line 60: Highest is usually used to refer to a certain factor. Do you mean higher? Please confirm the highest or higher, which word you want to express in this article.

Method

Ÿ   Line 77-79: The purpose, scoring method, range, cut point of clinical frailty scale, MMSE, and HDS-R are unclear. Please provide an additional explanation for this part.

Ÿ   2.2 Procedure: How many days are recorded for each participant of their TS, HGS, and oral mucosal moisture? The amount of data affects the reliability of the analysis results and it is necessary to explain.

Ÿ   Line 104-105: As I know, both TP and HGS are measuring the maximum value. In the text, the maximum value of HGS was recorded. If the measure of interest is maximum TP, why average the three trials? The maximum value from the three trials should be used. In fact, it usually takes 3 trials to obtain a valid maximum TP. This problem may also account for the lower than expected TP values.

Ÿ   Line 121: Why use median and interquartile range (IQR) to present numeric data of TS, oral mucosal moisture, and HGS? Is there a special purpose? Furthermore, the median is used to represent the degree of dispersion and is not suitable for calculations.

Results

Ÿ   Line 130-134: It is recommended to move this paragraph to the materials and methods section, and revised the repeated sentences about participants.

Ÿ   Line 137-138: Please explain the meaning of the numbers in brackets after MMSE and HDS-R.

Ÿ   Line 140-145: Regarding the description of the results in Lines 140-150, it is recommended to replace them with a table to increase the readability of the readers.

Ÿ   Table1: Please confirm the statistical analysis method used and that the results presented in Table 1 are from linear regression analysis? The results of linear regression analysis and logistic regression analysis are presented in different ways. Table 1 looks more like the results of the logistic regression analysis. Furthermore, the advantage of using multivariable-level cannot be seen from this result table. Please present the statistical results in the correct way.

Discussion

Ÿ   In Line 171: TS was 13.7% 171 higher after lunch, please provide relevant data in the result table or explain how it was calculated?

Other comments

Ÿ   Did the researchers record the participants' activity and types of diet? Previous research has pointed out that food texture is related to TS. Participants' activity and texture of diets are likely to affect or motivate tongue strength performance. How did the authors rule out this part of the effect?

Reviewer 3 Report

Comments and Suggestions for Authors

This is an interesting study that investigated changes in tongue muscle strength during the day in frail elderly people. However, there are parts that need to be corrected and supplemented, and parts that need to be checked in detail.

Page 2

Materials and Methods

1. inclusion criteria: (i) able to follow instructions: In this study, the cognitive function of the subjects was evaluated using an assessment tool such as MMSE. Therefore, it would be better to set the standard based on specific MMSE scores rather than the condition of "can follow instructions".

2. Procedure: participants were asked to place the balloon in their mouth: It would be better if an explanation of the anatomical position in the oral cavity where the bulb is placed when measuring tongue pressure is added.

Page 3

3. Line 103: push the balloon with their tongue against their hard palate: Which part of the tongue was instructed to press the balloon?

4. Line 104: for 7 seconds: Is there a reason for holding the maximum pressure for 7 seconds? 7 seconds is quite a long time.

5. Line 106-109: Please explain the hand grip strength measurement posture and maximum grip holding time (When measuring tongue pressure, the maximum pressure was maintained for 7 seconds).

6. Tongue pressure was measured three times and the average value was used for analysis, and grip strength was measured twice and the maximum value was used for analysis. Can you explain the reason for these difference?

7. line 112-113, oral moisture-checking device: Isn't this the product name? Please provide the name of the actual product.

Page 4

8. line124-125: The Kruskal-Wallis test is a non-parametric test method that compares the sizes of three or more independent groups. Friedman test should be used when non-parametric analysis of repeated measurement data for the same subjects. It seems that the statistical method should be changed and analyzed again.

9. line 126-127: multivariable-level linear regression analysis: Regression analysis is a statistical technique used to establish a causal relationship between an independent variable and a dependent variable. What variables did you try to establish a causal relationship with? If you are sure that you have used an appropriate statistical method, please describe your method of analysis in more detail.

Results

10. line131-133: Only those who were able to follow instructions were included in the study, and an explanation is needed as to why they were unable to follow instructions.

11. Line 135 Table 1: Table 1, which contains demographic information, is nowhere to be seen. Table 1 at the bottom of page 4 appears to be a table showing the results of the regression analysis.

12. Has the reliability (Intra-rater reliability, inter-rater reliability) of the measurement been evaluated?

13. The median values of tongue pressure measured in the morning/noon/evening were 20.8, 20.7, and 20.0, respectively. As a result of the regression analysis, you mentioned that a significant difference was confirmed between morning and noon. Does this seem to be a clinically meaningful difference?

Page 5

14. Line 171, TS was 13.7% higher after lunch: It is necessary to explain how the value of 13.7% was calculated by the formula. 20.7/20.8*100=13.7 ?

15. line 174, durability? Do you mean endurance?

Round 2

Reviewer 1 Report

Comments and Suggestions for Authors

Thank you for carefully and thoughtfully considering my comments on the original submission.  This version is improved in its more cautious treatment of the results and overall interpretation of the study.

I understand the method and the revisions do not change that understanding. I also understand that these authors have used this "squeeze and hold" method and have averaged data in past publications. It may be a reliable measure (line 121), but that doesn’t mean it’s valid.  I still contend that holding maximum voluntary contraction for 7 seconds x 3 trials can fatigue the tongue, and that the maximum value should be used for tongue strength assessment, not the average.  It is possible that I don't understand the instrument used (why would it take 7 sec to stabilize?). This sounds like a asymptotic measure after the peak value is reached and the tongue begins to fatigue.  The Editor may decide that this method is acceptable, but I think it would benefit readers to know that this is not a typical way of evaluating tongue strength and that it may underestimate maximum tongue strength. Fortunately, the method was used consistently throughout the study, so any reduction in maximum tongue pressure exertion may be similarly affected for all timepoints (that is, if one can assume a linear relationship between maximum pressure and asymptotic pressure generation with a sustained task).

Comments on the Quality of English Language

Some phrasing is awkward, but it is quite interpretable.  I do recommend changing the words "wetting level" on line 56 to "lubrication" for example.

The revision regarding the experience of the dentist and speech therapist did not improve meaning. The dentist may have experience as a dentist, but the speech therapist would be the only one with experience diagnosing dysphagia.  Please clarify. This may require something as simple as inserting the word "a" before "speech therapist" in line 92.

Fig 2 appears to be a screenshot that accidentally captured “Plot Area” in the image.  Please correct.  I'm glad to see the addition of the figures.

Author Response

Responses to the Reviewers' Comments

We greatly appreciate the efforts of the reviewers for their constructive comments in evaluating our manuscript. Each comment has been carefully considered, and responses have been provided for each point. All the revisions in the manuscript are highlighted in yellow. Our responses to the reviewers and the changes made in the revised manuscript are as follows.

Reviewer 1 comments:

Point1: Thank you for carefully and thoughtfully considering my comments on the original submission.  This version is improved in its more cautious treatment of the results and overall interpretation of the study.

I understand the method and the revisions do not change that understanding. I also understand that these authors have used this "squeeze and hold" method and have averaged data in past publications. It may be a reliable measure (line 121), but that doesn’t mean it’s valid.  I still contend that holding maximum voluntary contraction for 7 seconds x 3 trials can fatigue the tongue, and that the maximum value should be used for tongue strength assessment, not the average.  It is possible that I don't understand the instrument used (why would it take 7 sec to stabilize?). This sounds like a asymptotic measure after the peak value is reached and the tongue begins to fatigue.  The Editor may decide that this method is acceptable, but I think it would benefit readers to know that this is not a typical way of evaluating tongue strength and that it may underestimate maximum tongue strength. Fortunately, the method was used consistently throughout the study, so any reduction in maximum tongue pressure exertion may be similarly affected for all timepoints (that is, if one can assume a linear relationship between maximum pressure and asymptotic pressure generation with a sustained task).

Response 1:  As you rightly noted, having the tongue pressed against the hard palate for roughly 7 seconds and repeating this thrice could lead to tongue fatigue, potentially leading to an underestimation of the maximum TS. I have included this point in the limitations section.

Discussion:

“In this study, the measurement of TS took approximately 7 seconds, consistent with that in previous studies [7,18]. We used the average of three measurements. However, there is potential for tongue fatigue to set in, which may affect the results. Additionally, a discrepancy was noted between the maximum value of TS and our measurements.” (Page 7, Lines 267-271)

Point2: Some phrasing is awkward, but it is quite interpretable.  I do recommend changing the words "wetting level" on line 56 to "lubrication" for example.

The revision regarding the experience of the dentist and speech therapist did not improve meaning. The dentist may have experience as a dentist, but the speech therapist would be the only one with experience diagnosing dysphagia.  Please clarify. This may require something as simple as inserting the word "a" before "speech therapist" in line 92.

Fig 2 appears to be a screenshot that accidentally captured “Plot Area” in the image.  Please correct.  I'm glad to see the addition of the figures.

Response 2:  Thank you for your advice. For this revision, the change has been made, and the English text has been proofread by a native speaker once again.

To clarify, dentists and speech therapists actively participate in dysphagia rehabilitation in the Japanese medical context. The dentist involved in this study, similar to the speech therapist, has over 20 years of experience in evaluating and rehabilitating dysphagia. In Japan, dentists frequently assess dysphagia through videoendoscopic and videofluoroscopic examination of swallowing. They collaborate closely with multidisciplinary teams to provide comprehensive dysphagia rehabilitation. We hope this provides a clearer understanding of the roles of both professionals in Japan in our study.

In addition, we have inserted the original data for Figure 2.

Reviewer 2 Report

Comments and Suggestions for Authors

I am satisfied with the authors response to previous comments. 

Author Response

Thank you very much for allowing us to make this paper better.

Reviewer 3 Report

Comments and Suggestions for Authors

Thank you for your revision.

Author Response

(The authors gave the same response as above.)
